# Modification of Liquid Separation Membranes Using Multidimensional Nanomaterials: Revealing the Roles of Dimension Based on Classical Titanium Dioxide

**DOI:** 10.3390/nano13030448

**Published:** 2023-01-21

**Authors:** Pei Sean Goh, Zahra Samavati, Ahmad Fauzi Ismail, Be Cheer Ng, Mohd Sohaimi Abdullah, Nidal Hilal

**Affiliations:** 1Advanced Membrane Technology Research Centre, Faculty of Chemical and Energy Engineering, Universiti Teknologi Malaysia, Johor Bahru 81310, Malaysia; 2NYUAD Water Research Center, New York University Abu Dhabi, Abu Dhabi 129188, United Arab Emirates

**Keywords:** nanocomposite membrane, titanium oxide, liquid separation, membrane modification, nanomaterial dimensions

## Abstract

Membrane technology has become increasingly popular and important for separation processes in industries, as well as for desalination and wastewater treatment. Over the last decade, the merger of nanotechnology and membrane technology in the development of nanocomposite membranes has emerged as a rapidly expanding research area. The key motivation driving the development of nanocomposite membranes is the pursuit of high-performance liquid separation membranes that can address the bottlenecks of conventionally used polymeric membranes. Nanostructured materials in the form of zero to three-dimensions exhibit unique dimension-dependent morphology and topology that have triggered considerable attention in various fields. While the surface hydrophilicity, antibacterial, and photocatalytic properties of TiO_2_ are particularly attractive for liquid separation membranes, the geometry-dependent properties of the nanocomposite membrane can be further fine-tuned by selecting the nanostructures with the right dimension. This review aims to provide an overview and comments on the state-of-the-art modifications of liquid separation membrane using TiO_2_ as a classical example of multidimensional nanomaterials. The performances of TiO_2_-incorporated nanocomposite membranes are discussed with attention placed on the special features rendered by their structures and dimensions. The innovations and breakthroughs made in the synthesis and modifications of structure-controlled TiO_2_ and its composites have enabled fascinating and advantageous properties for the development of high-performance nanocomposite membranes for liquid separation.

## 1. Introduction

Membrane-based liquid separation processes have substantial roles in industries such as food processing, pharmaceuticals, and petrochemicals [1,2,3,4,5,6]. Over the last two decades, membrane technology has also been increasingly employed to address water scarcity issues through wastewater reclamation and desalination [7,8,9]. In membrane separation, a semi-permeable membrane serves as a barrier to selectively allow the transport of some molecular or ionic species while hindering others. Compared to many traditionally used separation processes such as distillation, membrane separation can be performed at ambient temperature without the presence of chemicals. The ease of up-scaling and integration with other separation or treatment processes makes membrane separation grow in demand [10]. In general, membrane processes can be accomplished through several means, which mainly differ in the driving forces involved in the processes. Pressure-driven membrane processes like microfiltration (MF), ultrafiltration (UF), nanofiltration (NF), and reverse osmosis (RO) separate mixtures mainly based on size exclusion in the presence of external hydraulic pressure [11]. Pressure-driven membrane processes are matured technologies that have been well-established. Currently, they have been widely explored for desalination and wastewater treatment because of their reliability in delivering treated water with desired quality for both reuse and discharge [12]. More recently, osmotically driven membrane processes such as forward osmosis (FO) and pressure-retarded osmosis have attracted attention owing to their great potential to perform separation in a more sustainable manner as osmotic gradient can be used as a driving force instead of externally applied pressure [13,14]. Without the requirement of high pressure in their operations, FO has been actively investigated as an attractive alternative to pressure-driven processes such as NF and RO [15]. Other useful membrane-based liquid separation processes such as pervaporation [16], membrane distillation [17], and electrodialysis [18] play equally important roles in a wide range of separation processes.

The membrane is the most important component that dictates the performance and efficiency of a membrane filtration process. A high-performance membrane warrants the high productivity of the separation process and the high purity of the separated stream, while exhibiting commercially attractive characteristics such as being durable and reproducible. In liquid separation such as desalination and wastewater treatment, maintaining a good balance between water permeability and solute rejection ability is of utmost importance to ensure the practicability of the separation process. Furthermore, the resistance of membranes toward various types of fouling and chlorine attacks is also a major concern in membrane development. While commercially available membranes are still largely based on conventional polymeric materials such as polysulfone (PSF) and poly(vinylidene fluoride) (PVDF), innovations have been initiated in laboratory-scale studies to overcome the limitations of these conventional membranes. Numerous approaches have been considered, from exploring new monomers or polymeric materials [19,20], tailoring membrane structure and layers [21], and modifying membrane surface through introduction of functional compounds to incorporating newly developed additives within the polymeric membranes [22,23,24,25]. Nanocomposite membranes incorporated with a broad spectrum of nanomaterials have attracted increasing interest on account of their desired characteristics. As its name implies, the nanocomposite membrane is a hybrid membrane formed when nanomaterials are introduced to a polymeric material that acts as a host matrix. The nanomaterials serve as dispersing additives or nanofillers to render their unique nanoscale properties to the bulk polymer. This integrated membrane combines the advantages of both entities to achieve synergistic effects that are not seen when nanomaterials and polymeric membranes are used independently [26]. Known as new-generation membranes, nanocomposite membranes enhance separation performances to address intrinsic issues of polymeric membranes. They represent a new strategy for improving the membrane performance in terms of permeability, selectivity, anti-fouling, chlorine resistance, and long-term stability. The improvements are materialized by altering the physical and chemical characteristics of the membrane, such as the textural properties like surface porosity, as well as surface chemistry like wetting ability and charges [27]. As such, the drawbacks of conventional membranes derived from commercial polymers especially in terms of permeability–rejection trade-off and high fouling propensity can be addressed with the new properties introduced by the nanomaterials.

Thousands of nanomaterials have been synthesized and explored for the preparation of nanocomposite membranes for liquid separation. Although criteria have not been systematically set for the selection of nanomaterials, it is generally agreed that the nanomaterials used for the preparation of nanocomposite membranes should have a suitable size and length so that the overall structural integrity, especially the thin selective layer of the membrane that is responsible to solute the rejection ability, is not disrupted. It is also desirable to ensure good interfacial interaction between inorganic nanomaterials and the polymeric matrix. The presence of functional groups on the surface of nanomaterials is beneficial to improve their interactions with the polymer chain, hence promoting their dispersibility and distributions within the nanocomposite membranes. The earlier attempts to fabricate nanocomposite membranes were mainly focused on the use of metal and metal oxide nanoparticles as these nanomaterials are widely available or can be synthesized in the laboratory through well-understood reactions. Cu, Ag, TiO_2_, ZnO, and CuO, just to name a few, have been explored as functional additives for nanocomposite membranes [28]. Carbon nanostructures that include various dimensional allotropes of carbon family such as fullerene, carbon nanotubes, and graphene later emerged as attractive candidates for the development of nanocomposite membranes [29,30]. These carbon-based nanomaterials exhibit many desirable properties that can be utilized to enhance water separation, such as ultrafast water transport and antimicrobial properties. Metal organic frameworks [31] and layered double hydroxide [32] have also become popular for the preparation of nanocomposite membranes owing to their cage-like or layered structures that can offer high surface functionality, high chemical tunability, and large surface area.

Thanks to the countless efforts made in advancing material science and engineering, constant improvements and breakthroughs have been made in synthesizing and characterizing new nanomaterials. The manipulations and fine tuning of the synthesis conditions enables the materials of the same elemental compositions to be architectured into various structures and dimensions. This possibility results in a series of fascinating nanomaterials that exhibit structural advantages that match the expectations. For nanomaterials used for liquid-related applications, their geometrical structures and dimensions can lead to special functions and roles. Tailoring the geometries of nanomaterials allows for more flexibility than simply changing their sizes as additional properties and functions [33,34]. For example, one-dimensional (1D) tubular-structured nanomaterial can be advantageously used as a water channel to facilitate water flow [35], but this feature is unmet by zero-dimensional (0D) spherical nanomaterial made up of the same chemical composition. On the other hand, two-dimensional (2D) nanomaterials enable highly anisotropic physical and chemical properties that are not found in zero- or 1D nanostructures [36]. When multidimensional nanomaterials are used as the inorganic fillers for the fabrication of nanocomposite membranes, this forms an important basis for the selection of nanomaterials with the right geometrical structure to cater to their intended application.

A significant amount of work related to the development of membranes for liquid separation processes have been published in the past 10 years, indicating the importance of this topic to industries and communities. Membrane modification through the introduction of nanomaterials is one of the most extensively reviewed topics in view of the effectiveness of this strategy in tailoring the physico-chemical properties of membranes [37,38,39,40,41]. Significant efforts have been made in developing nanocomposite membranes using metal oxides including TiO_2_, which is well-known for its low cost, abundancy, high chemical stability, and nontoxicity to the environment and humans. Regardless of the types and classes of the nanomaterials for nanocomposite membrane preparation, it is imperative to understand the structural characteristics of nanomaterials in different dimensions so that the dimension-dependent features can be fully exploited. Nevertheless, current reviews in this area are not focused on this aspect and some research questions remain to be addressed. Can all geometrically different nanostructures be suitably used for the preparation of nanocomposite membranes? What are the transport behaviors of water molecules when passing through nanofiller with multidimensions in the nanocomposite membranes? What are the orientations of nanofiller with multidimensions when they are dispersed within the polymeric matrix of nanocomposite membranes? Based on TiO_2_, which known as one of the most classical examples of multidimensional nanomaterials used for the development of nanocomposite membranes, this review aims to provide an overview on the roles of multidimensional nanomaterials in the state-of-the-art modifications of liquid separation membrane. The novel syntheses of shape-controlled TiO_2_ in various dimensions and morphology have led to many interesting environmental and energy applications. The dimension-tailored TiO_2_ nanostructures have been used to prepare liquid-separation nanocomposite membranes that demonstrate improved separation properties in terms of water flux and solute rejection, as well as other desired characteristics such as mechanical strength and fouling resistance. The surface hydrophilicity and antibacterial and photocatalytic properties of TiO_2_ are particularly attractive and important for liquid separation membranes used for water and wastewater treatment. The potentials and efficiencies of TiO_2_-based photocatalytic nanocomposite membranes for the removal of pharmaceutical products have been reviewed [42]. The degradation mechanisms and the optimum conditions of the photoreactor have been summarized. The development of a nanocomposite membrane based on the hybrid of TiO_2_ and graphene oxide (GO) for various applications has also been discussed. The use of GO associated with TiO_2_ addresses the limitations of TiO_2_, leading to synergistic effects that can heighten the performance of TiO_2_ and the resultant nanocomposite membranes.

Instead of elaborating the extensive work made in TiO_2_-based nanocomposite membranes for liquid separation, this review primarily aims to identify the knowledge gaps in the design of a high-performance nanocomposite membrane through the selection of TiO_2_ nanomaterials with the right material dimension and structures. In the following sections, the synthesis approaches and features of multidimensional TiO_2_ are first reviewed, followed by the current strategies used to introduce TiO_2_ into various kinds of liquid separation membranes. The performances of TiO_2_-incorporated nanocomposite membranes are discussed with attention placed on the special features rendered by their structure and dimensions. The challenges are highlighted and some recommendations for future studies to address the bottlenecks in this field are provided accordingly. By identifying the structural features and advantages demonstrated by nanomaterials that can be synthesized in multidimensions, the most appropriate structure and dimension can be selected. This allows the chosen nanomaterial to fit the configuration of membranes and the intended applications of the resultant nanocomposite membranes. This serves as a useful proposition to maximize the potential of nanocomposite membranes.

## 2. Features of Multidimensional TiO_2_ for Liquid Separation Membrane

Owing to its many commercially attractive characteristics, TiO_2_ has become a promising material for industrial application. TiO_2_ pigment is one of the most commercially important materials since millions of metric tons of TiO_2_ pigment have been commercialized annually worldwide [43]. TiO_2_ also has an important position in the field of water treatment and reclamation, especially for the removal of various hazardous water contaminants and pathogens from polluted water bodies [44,45,46]. TiO_2_ can be synthesized into different morphologies that represent different physicochemical properties such as shape, surface areas, and crystal orientation, which can cater to a variety of applications. For TiO_2_ nanostructures in the size range of 1–100 nm, three geometric structures that are 0D spherical, 1D elongated, 2D planar, and 3D assembly structures have been synthesized. TiO_2_ nanostructures in various dimensions have been comprehensively reviewed in terms of their syntheses and modifications, morphologies and structural properties, as well as applications [47,48,49,50,51]. Figure 1 shows the microscopic images of TiO_2_ of different geometric structures and dimensions. Spherical TiO_2_ has been synthesized in the form of nanospheres [52], nanoplates [53], nanopores [54], and quantum dots [55]. A nanosphere is defined as a homogenous matrix system, while quantum dots are nanocrystals of semiconductors that are small enough to exhibit quantum mechanical properties [56]. Elongated TiO_2_ can be synthesized as hollow structures like nanotubes or non-hollow solid structures like nanorods [57], nanofibers [58], and nanowires [59]. Although some of these structures share the same synthesis approaches, they may be slightly different in terms of their physical properties. Both nanorods and nanofibers can be formed by growing the nuclei of a TiO_2_ precursor in vertical or horizontal directions. Nevertheless, nanofibers tend to be flexible and have a higher-density structure, while nanorods possess higher rigidity in nature [60]. While both nanorods and nanowires have a similar morphology, the former has lengths that typically fall in the range of 10–120 nm, while the latter has lengths that are not delimited and are much longer than their diameters [61]. On the other hand, compared to nanorods, nanotubes have a more complex hollowed structure. A TiO_2_ nanotube possesses a large surface area that is related to its internal and external surfaces [62,63], while a TiO_2_ nanorod exhibits a quantum-confined effect that is beneficial for improving photoinduced charge transport efficiency and charge carrier separation rate [64]. Planar or sheet-like TiO_2_ in the forms of nanosheets [65], nanoplates [66], and nanofilms [67] have been synthesized. The thin atomic thickness and large planar size endow 2D TiO_2_ with a high specific surface area and large surface-to-volume ratio [68]. Three-dimensional structures of TiO_2_ normally consist of arrays and assemblies such as nanoflowers [69] and nanoforests [70]. While reserving the unique properties at nanometer scale, the 3D TiO_2_ hierarchical structure micrometer-scale can be handled more easily [71]. Compared to TiO_2_ nanoparticles, which normally suffer from a severe agglomeration issue, 3D microscale TiO_2_ nanostructures have lower agglomeration tendency and are easier to be isolated from the suspension [72]. The most frequently used synthesis techniques for these 0D to 3D TiO_2_ nanostructures are the hydrothermal and sol–gel methods using commercial TiO_2_ nanoparticles or titanium solutions such as titanium isopropoxide (TTIP) and tetrabutyl titanate (TBOT) as precursors [73].

As morphologies have immense effects on the optical, electronic, and catalytic properties, the applications of TiO_2_ are closely associated to its morphology and dimensions. One-dimensional spherical TiO_2_ has been conventionally used for electronics, sensors, and antimicrobial applications [74,75]. Tubular TiO_2_ has served as an emerging candidate for electrode materials used in electrochemical microdevices, which hold potential for energy delivery or harvesting in biological and medical fields [76]. Several studies have shown that TiO_2_ nanomaterials with nanosheet structures can offer improved photocatalytic properties compared to their spherical and tubular counterparts [77,78,79]. On the other hand, a well-controlled 3D TiO_2_ assembly demonstrated lower reflectance, higher surface area, and more effective light harvesting for better photocatalytic activity [72]. The control of the distribution of nanomaterials within the polymer matrix is an important yet challenging task in the preparation of nanocomposite membranes. Unlike 0D nanostructures with isotropic behavior, the uniform alignment of 1D and 2D nanofillers in the desired direction offer an opportunity to exploit their anisotropic features that are advantageous for liquid separation membranes [80]. It has been demonstrated that the optical properties and the photocatalytic activity of TiO_2_ can be enhanced by reducing the dimension from 3D bulk to 2D nanosheet along the {001} and {101} planes [81]. Under controlled synthesis conditions, the geometry and morphology of nanostructures can be interconverted into other nanostructures with the same chemical composition but lower or higher dimension. For instance, it has been demonstrated that TiO_2_ nanosheets can be converted into hierarchical hollow hydrous TiO_2_ microspheres through a mild one-pot template-free hydrothermal condition [82]. This architecture combines the unique advantages derived from nanosheets and hollow structures, which are beneficial for many applications.

Through the control of their structures, size, and functionalities, TiO_2_ nanomaterials have led to innovations in many applications [83,84,85]. As one of the most attempted nanomaterials for nanocomposite membrane preparation, TiO_2_ possesses characteristics that are highly desired for liquid separation membranes. The incorporation of TiO_2_ with many interesting surface chemistries and structural properties expands the potentials of nanocomposite membranes to perform challenging separations. For instance, the treatment of wastewater containing emerging contaminants such as pharmaceutically active compounds, endocrine disruptive chemicals, and other organic micropollutants is made possible by integrating the separation efficiency of polymeric material and the newly added functionalities of TiO_2_. Many polymeric membranes used for liquid separation are commercially available. However, due to their hydrophobicity, polymeric membranes such as PSF and PVDF are prone to membrane fouling. Therefore, the surface hydrophilicity of TiO_2_ has been advantageously harnessed to improve the water permeability and antifouling properties of the nanocomposite membranes. The high hydrophilicity of TiO_2_ can be exploited in the design of nanocomposite membranes with improved water flux and antifouling properties. When incorporated into the polymeric matrix, the hydroxyl groups attached on the surface of TiO_2_ render improved hydrophilicity of the resultant nanocomposite membranes [86]. Surface hydrophilicity of a liquid separation membrane has been closely related to water permeability in which hydrophilic surface could attract and promote the transport of water molecules across the membranes [87]. Additionally, the hydroxyl groups also facilitate the formation of a hydration layer, which can act as a barrier for the deposition of organic foulants. Organic fouling is known as the most dominant fouling that takes place in liquid separation, especially in feed water containing high amounts of natural organic matter [88,89]. Therefore, improving the surface hydrophilicity is an effective way to suppress organic fouling [90,91].

Under UV light irradiation, the strong oxidative effect of TiO_2_ can be initiated and utilized as a photocatalytic disinfectant. TiO_2_ produces oxidative free radicals that exhibit bactericidal and antiviral properties against several types of microorganisms including Gram-negative and Gram-positive bacteria and fungi. The interaction between the TiO_2_ and bacteria such *E. coli*, and the subsequence bacterial growth inhibition and killing action have been well-discussed [92,93]. It has also been evidenced that the attachment of a TiO_2_ nanoparticle to the *E. coli* cell wall led to bacterial loss of cultivability in the dark because of the impairment of the integrity of the cell wall membrane [94]. The introduction of TiO_2_ as a biocidal agent in a liquid nanocomposite membrane has enabled a new antibiofouling function during liquid separation. Interestingly, the TiO_2_-incorporated nanocomposite membrane can exert a non-contact biocidal action to achieve disinfection capabilities; thus, the release of potentially toxic nanoparticles is not of major concern. The antimicrobial efficiency of TiO_2_ is affected by many factors including their crystal phase, structure, size, specific surface area, and adsorption nature [95]. TiO_2_ nanotubes in anatase phase exhibited the highest antibacterial activity compared to their rutile and amorphous counterparts [96]. The antibacterial behavior of TiO_2_ is influenced by its diameter, but independent of its length. Based on the antibacterial mechanisms of TiO_2_ nanotube arrays, Ji et al. observed an antibacterial effect of TiO_2_ nanotubes, which was related to the surface free energy and nano-topography [97]. The rupture and the subsequent death of *E. coli* were mainly attributed to the destructive effect of the nanotube geometry on the cell membrane.

TiO_2_ has been widely used as a photocatalyst in a variety of applications and products related to the environment and energy owing to its promising electronic properties that offer high photoactivity to effectively transform hazardous materials into less hazardous compounds. Interestingly, when used as an inorganic filler of nanocomposite membranes, TiO_2_ can also confer photocatalytic activity as a new functionality to the membranes to enable photocatalytic removal of a wide range of organic compounds [98]. In the so-called photocatalytic nanocomposite membranes, the integration of photocatalyst and membrane provides an effective treatment of wastewater because of the dual action on the organic pollutants, namely photodegradation and solute rejection [99]. During filtration, organic pollutants such as dyes and endocrine-disrupting compounds (EDCs) are photodegraded by TiO_2_ upon the absorption of visible or UV light by the nanocomposite membrane, which leads to the creation of different reactive oxygen species. In the photocatalytic membrane reactor, filtration via size exclusion takes place simultaneously through the selective layer of membrane to produce treated water. The presence of a TiO_2_ photocatalyst allows the breakdown of complex organic pollutants into simpler and smaller substances, hence reducing the fouling tendency. As the photodegraded products are often harmless or less harmful than the original compounds, the disposal of retentate produced from the filtration creates fewer environmental issues. The potential of TiO_2_ in mitigating the fouling of photocatalytic membranes has been discussed based on several aspects such as the composition of the photocatalytic membrane and the operating conditions of the photocatalytic membrane reactor [100]. Recently, the development of 1D TiO_2_ nanotubes and their hybridization with metal, metal oxides, and other organic or inorganic composites has become an emerging field in photocatalysis as the structures have been associated with excellent nanocomposite sensitization in the range of visible light and large surface-active sites [101].

## 3. Strategies to Introduce TiO_2_ into Liquid Separation Membranes

Liquid separation membranes can be fabricated in different configurations and structures, depending on the intended membrane processes and applications. Conventional pressure-driven membrane processes such as MF and UF are mainly based on integrally skinned asymmetric membranes that are fabricated by the phase inversion technique. The membrane consists of a thin skin layer and a porous substructure that are formed simultaneously during the precipitation process. The liquid separation nanocomposite membrane fabricated through phase inversion is mainly made from PSF and PVDF as the host matrix. Other fabrication techniques such as melt blending have also been used to form nanocomposite membranes with low-density polyethylene as the base polymer [102]. A photocatalytic nanocomposite membrane based on ultra-high molecular weight polyethylene (UHMWPE) that was melt-blended with up to 80 wt% of TiO_2_ nanoparticles has been reported [103]. Thin-film composite (TFC) membranes, which comprise ultra-thin selective layers formed over a microporous polymeric support membrane, is the most widely used membrane structure for RO and FO. A polyamide selective layer is commonly formed through interfacial polymerization of two monomers in aqueous and organic phases. The primary advantage of TFC membranes over their integrally skinned asymmetric counterpart is their ability to demonstrate greater water flux and selectivity. However, the polyamide layer has a lower resistance toward oxidants and chlorine chemicals. In addition, the layered structure of TFC membranes allows greater freedom in the design of membranes as the substrate layer and selective layer and can be independently modified and optimized using different modification approaches and modifying agents. The incorporation of nanomaterials, which is normally expensive and produced in small quantity, can be concentrated in the thin selective layer, instead of the entire membrane structure. While minimizing the wastage of nanomaterials, this strategy also can maximize the exposure of nanomaterials at the membrane surfaces in contact with feedwater during the filtration process. A similar concept has been applied for dual-layer hollow-fiber nanocomposite membranes, which consist of inner and outer layers of different compositions co-extruded simultaneously [104,105]. The nanofiller can be selectively introduced in the inner or outer layer of the dual-layer nanocomposite membrane, depending on the flow direction of the feedwater during filtration.

TiO_2_ can be introduced through multiple ways to the polymeric material to form a nanocomposite membrane. As illustrated in Figure 2, three major approaches, namely (i) ex situ incorporation, (ii) in situ formation, and (iii) post-membrane fabrication surface coating or grafting have been established for this purpose. Currently, ex situ incorporation is the most common method used to prepare a liquid separation nanocomposite membrane. TiO_2_ of desired structures and dimensions is first synthesized and optimized to obtain the desired properties. Subsequently, the TiO_2_ is introduced into the polymer dope to form a stable suspension prior to membrane formation. In the case of TFC membranes, the TiO_2_ nanofiller can be introduced into the monomers in organic or aqueous phase prior to interfacial polymerization of the selective layer. It is generally agreed that the dispersion of hydrophilic spherical nanoparticles in hydrophobic polymeric phase remains challenging. In addition to the inorganic–organic compatibility, the loading and size of TiO_2_ also affect their dispersion. A surface modification of the nanofiller through functionalization using bridging agents or surface compatilizers such as silane [106,107], amine [108], and polydopamine [109,110,111] are often required to improve the dispersion. Some of the expected physical properties of nanocomposite membrane such as mechanical properties are highly dependent on the uniformity of nanofiller dispersion as the agglomerations of nanofillers reduce the load transfer efficiency between the nanofiller and organic phases [112]. In the context of liquid separation, which involves mass transport, the formation of unfavorable voids around the agglomerated nanofiller results in non-selective transport of both desired and undesired substrates across the membrane. When this phenomenon takes place, the water permeability is drastically improved at the expense of selectivity.

The in situ formation of inorganic nanofiller within the polymer host matrix is known as a promising strategy to improve nanomaterial dispersion and resolve the agglomeration issue [113]. High loading of nanofillers can be achieved with better nanofiller dispersion and distribution. Among the synthesis methods of metal oxide, the sol–gel method is particularly appropriate for the in situ formation nanofillers in organic polymer given the simple and mild conditions of sol–gel chemistry such that the properties of the polymer can be preserved to the maximum extent [114]. The nature of precursor and solvent are important parameters in sol–gel synthesis of TiO_2_ as they affect the crystallization and in turn the surface area of TiO_2_ [115]. In situ formation of TiO_2_ can be done by the blending of TiO_2_ sol with polymer dope, followed by synchronous sol–gel and phase-inversion precipitation processes to form the nanocomposite membranes. The in situ transformation of sol–gel in the mixture of TiO_2_ sol, PVDF and poly(acrylic acid) (PAA) has been reported [116]. As shown in Figure 3a, the polymer chains coordinated with the TiO_2_ sols resulted from the partially hydrolyzed tetrabutyl titanate in the sol–polymer dope. The self-assembly of TiO_2_ sol at the membrane surface and pores took place, and gel nanoparticles were simultaneously formed through the polycondensation of the TiO_2_ sol. To harness the photocatalytic activity of TiO_2_, the amorphous must be transformed into a photocatalytically active anatase crystalline phase [117].

Some applications of nanocomposite membranes require the maximum exposure of the nanomaterials to optimize the performance [119]. When the nanomaterials are embedded within the polymeric matrix, some of the surface-related functionalities are shielded or hampered. For instance, the active sites of nanomaterials with photocatalytic ability are partially covered and not fully accessed by light for activation, thus deteriorating the photocatalytic activity. Most antimicrobial activities of nanomaterials are through the contact killing of microorganisms. When they are incorporated within the polymer matrix, the embedded nanomaterials are constrained from the contact-killing action; hence, the antimicrobial capability is also reduced. To address these issues, nanocomposite membranes have been formed by introducing the nanomaterials onto the surface of an as-fabricated polymeric membrane. By comparing photocatalytic nanocomposite membranes prepared from the three major fabrication approaches, Tian et al. observed the highest degradation efficiency and most improved recyclability when TiO_2_ nanoparticles were deposited on the membrane surface [120]. Especially, the deposition of multidimensional TiO_2_ on the membrane surface can be accomplished by physical and chemical means. The physical method involves the deposition of TiO_2_ through physical interactions, which can be accomplished by various types of substrate coating methods such as spin or dip coating and layer-by-layer assembly [121]. On the other hand, the chemical method involves the establishment of chemical bonding between the functional terminal groups of TiO_2_ and the polymer chain. Surface functionalization of TiO_2_ or membrane surface is usually required to activate their surfaces for the subsequent chemical interactions [122,123]. Graft polymerization of acrylic acid is one of the most-used strategies to functionalize the membrane surface prior to nanomaterial grafting [124,125]. Regardless of the types of interactions, it is crucial to form a thin and well-assembled TiO_2_ layer on the membrane surface to ensure good stability and to minimize the effect of resistance to water flow across the membrane. For a liquid separation membrane, the deposition of an additional layer can result in a drastic decrease in the water permeability. Although in most cases, the solute rejection ability can be improved by the additional layer, the permeability–rejection trade-off remains. Instead of using pre-formed TiO_2_, the formation of a TiO_2_ layer on the membrane surface can also be made through an in situ approach when the precursor of TiO_2_ is introduced onto the membrane surface, followed by hydrolysis and polycondensation or hydrothermal process [126]. The in situ growth of TiO_2_ has been performed on the surface of polymeric membranes through sol–gel method using titanium (IV) isopropoxide as precursor [118]. As illustrated in Figure 3b, the in situ hydrolysis of the TiO_2_ precursor allows the coordination between Ti^4+^ and the reactive sulfone groups and ether groups in the polymer backbone. The hydroxyl groups of TiO_2_ can also establish hydrogen bonding with these reactive functional groups. These interactions are favorable to achieve good TiO_2_ dispersion and firm attachment on the membrane surface.

When the geometries and dimensions of the nanomaterials are concerned, the abovementioned approaches come with advantages and limitations. Based on the state-of-the-art preparation approaches of nanocomposite membranes, ex situ introduction of nanofiller through direct blending with polymer dope prior to their formation is the most prevailing approach owing to its simplicity. It also allows high flexibility in optimizing the nanomaterials and polymeric host independently. The functionalization of these nanomaterials can be performed to enhance their dispersibility. For a nanocomposite membrane’s integrally skinned asymmetric structure, this approach is applicable to nanomaterials synthesized in various dimensions as the typical thicknesses of integrally skinned asymmetrics fall in the range 100–200 μm. As the TFC membrane consists of a thin selective layer with thicknesses of 100–300 nm, the embedment of vertically aligned rod-like or 3D nano-assemblies such as arrays with dimensions in the μm range is not favorable as they may protrude from the selective layer and destroy the layer. The suspension of a nanofiller precursor in polymer dope followed by the in situ growth simultaneously with membrane formation addresses the agglomeration issue of nanofillers. Nevertheless, this approach is currently restricted to the formation of spherical nanoparticles through mild conditions such as sol–gel chemistry. In situ growth of nanomaterials with more complex hierarchical structure in which their syntheses often call for multiple steps and extensive use of chemicals cannot be materialized in the polymer dope. Nanomaterial coating or grafting techniques provide the greatest flexibility to introduce multidimensional nanomaterials on the membrane surface while conserving the bulk properties of membranes. The major challenge in implementing this approach is to enable the desired functionalities of nanomaterials without jeopardizing the separation performance, especially water permeability.

## 4. Liquid Separation Membranes with Multidimensional TiO_2_: Performance Evaluation

Tremendous efforts have been made in developing nanocomposite membranes for various liquid separation applications, particularly for wastewater treatment and desalination. This section is organized to discuss the roles of multidimensional TiO_2_ in altering the physical properties and surface chemistry, thus the filtration associated performances of the nanocomposite membranes based on some representative studies.

### 4.1. One-Dimensional TiO_2_

The polyamide thin film of a TFC RO membrane has been incorporated with TiO_2_ nanotubes to address trade-off between water permeability and solute rejection of caffeine and bisphenol A as model endocrine-disrupting compounds [127]. As illustrated in Figure 4a, the water permeability enhanced by ~50% was attributed to the TiO_2_ nanotube-constructed nanochannels, which have provided additional water paths for water molecules to flow through. The oxygenated and hydroxyl functional groups attached onto the tubular surface of the TiO_2_ nanotube also played a role in increasing the water transport rate. In addition, the microvoids created at the interphase between the TiO_2_ nanotube and the polyamide matrix also further promote transporting water. Nevertheless, an excessive loading of TiO_2_ nanotubes higher than 0.01 wt% increased the thickness of the polyamide selective layer, which imposed additional mass-transfer resistance, hence reducing the water permeability. Khoo et al. prepared a TNT-incorporated NF membrane for desalination [128]. Prior to the incorporation, the TiO_2_ nanotubes were modified with methyl methacrylate to improve their dispersion in the polyamide matrix. The well-dispersed surface-functionalized TiO_2_ nanotubes facilitated the interfacial polymerization to form polyamide with a high crosslinking degree. The nanocomposite membrane showed enhanced water flux up to 16% without compromising NaCl rejection. However, as shown in Figure 4b, although the presence of TiO_2_ nanotubes has rendered antimicrobial properties to the thin film nanocomposite membrane against *E. coli* and *S. aureus*, the highest bacterial inhibition efficiency was exhibited by the nanocomposite membrane incorporated with unmodified TiO_2_ nanotubes. This indicates that the surface functionalization of TiO_2_ with a methyl methacrylate coating layer has hampered the antibacterial properties of the nanotubes.

### 4.2. Two-Dimensional TiO_2_

Through layer-by-layer dip coating, oppositely charged TiO_2_ nanosheets have been assembled on the surface of a polyamide RO membrane for oily wastewater treatment and desalination [129,130]. The surface charges of TiO_2_ were first altered by a simple immersion technique. The positively charged TiO_2_ nanosheets were obtained by protonating the as-synthesized TiO_2_ nanosheets in hydrochloric solution while the negative-charged TiO_2_ nanosheets were produced by immersing the as-synthesized nanosheets into aqueous tetrabutylammonium hydroxide. It was observed that the planar structures of TiO_2_ nanosheets facilitated the penetration of the nanosheets into the polyamide layer without noticeable agglomeration. The penetration was favorable to minimize the thickness of the TiO_2_ nanosheet coating layer, which could otherwise impose additional mass transport resistance. Together with the improved surface hydrophilicity, the laminated planar nanochannel played major roles in improving water transport efficiency while blocking larger species for improving solute rejection. In saltwater desalination, it was reported that the water permeability and NaCl rejection of the nanocomposite membrane were improved by 60% and ~2%, respectively, as compared to that of a neat TFC membrane.

### 4.3. Hybridized TiO_2_

Current research trends show that hybrid nanomaterials have emerged as a powerful tool to create multifunctional nanocomposite membranes [131]. The hybridization of TiO_2_ with other nanostructured materials has therefore become a topic of interest. The hybridization of TiO_2_ addresses the limitations, in which some of them have considerably impeded a more prolific use of the materials. In the hybridized system, TiO_2_ can serve in different ways, i.e., as a base for the deposition of other nanomaterials, as a deposited nanomaterial, and to couple with other nanomaterials. TiO_2_ hybridized with strong antimicrobial agents such as Ag nanoparticles demonstrated biocide features to render the nanocomposite membrane with improved antifouling properties toward microbes, including the commonly found *E. coli*. Ag-decorated TiO_2_ has been prepared by reducing silver nitrate by ascorbic acid on the surface of TiO_2_ nanoparticles [132]. The hybridized Ag/TiO_2_ was then introduced to the surface of a PVDF membrane that was pre-functionalized with 3-aminopropyl triethoxysilane. The resultant nanocomposite membrane demonstrated threefold improved pure water flux of 465.8 Lm^−2^h^−1^ and a higher *E. coli* killing rate than that of a neat PVDF membrane.

Kusworo et al. hybridized TiO_2_ and reduced GO (rGO) using a one-step hydrothermal method [133]. As shown in Figure 5ai, the 2D planar rGO nanosheets provided a site for the TiO_2_ nanoparticles to attach over the sheets. The rGO/TiO_2_ hybrid was introduced to PVDF dope to form a photocatalytic nanocomposite membrane via a phase-inversion technique. Besides increasing the surface hydrophilicity of the nanocomposite membrane, the TiO_2_ embedded in the PVDF matrix also facilitated the crosslinking of the PVDF chain with the surface-coated PVA layer through the interaction illustrated in Figure 5aii. With the incorporation of rGO/TiO_2_ of different loadings, the nanocomposite membrane exhibited photocatalytic activity that was increased by 30–100%. Similarly, Wu et al. prepared PVDF UF nanocomposite membranes using the hybrid of TiO_2_ and GO as nanofiller. The well-distributed TiO_2_/GO hybrid increased the surface hydrophilicity of the PVDF nanocomposite membrane, thus enhancing their antifouling properties [134]. By co-depositing the suspension of TiO_2_ and GO on the surface of acrylic-acid-grafted PVDF membranes, Tran et al. reported that the addition of a GO nanosheet to the TiO_2_ could reduce the aggregation of TiO_2_ nanoparticles, thus resulting in an enhancement of photocatalytic activity at low TiO_2_ loading under UV irradiation [135].

The coupling of TiO_2_ with metals, nonmetals, or other nanostructures also serves as a common solution to solve issues related to fast electron-hole recombination of the TiO_2_ photocatalyst. TiO_2_ nanotubes doped with boron have been used as nanofillers of photocatalytic dual-layer hollow-fiber membranes [138,139]. The selective embedment of TiO_2_ nanotubes in the outer layer of the dual-layer hollow-fiber membrane increased the surface hydrophilicity in a greater extent compared to integrally skinned asymmetric hollow-fiber membrane with random distribution of TiO_2_ nanotubes throughout the structure. The doping of boron improved the surface negativity of TiO_2_ nanotubes, which in turn facilitated the dispersion of the nanotubes across the membrane matrix. Hazaraimi et al. applied a facile chemical reduction method to deposit Ag nanoparticles on the external tubular surface of a TiO_2_ nanotube (Figure 5bi [136]. The hydrophilicity of TiO_2_ nanotubes influenced the thermodynamic stability of the polymer dope, hence altering the morphology of the nanocomposite membranes, where the spongy dense layer was shrunk as shown in Figure 5bii. The incorporation of Ag-doped TiO_2_ nanotubes into the outer layer of a dual-layer PVDF-matrix hollow fiber improved the pure water flux by >130% and color removal efficiency by 38% compared to the neat PVDF membrane. With the introduction of Ag nanoparticles through doping, the nanocomposite membrane also exhibited antibacterial efficiency of 95.8% against *P. aeruginosa.* TiO_2_ nanotubes synthesized by hydrothermal methods have been incorporated into PVDF to form a photocatalytic NF nanocomposite membrane for textile dye removal [137]. Prior to the addition to the PVDF dope solution, as schematically shown in Figure 5c, polyaniline (PANI) was in situ polymerized on the surface of the TiO_2_ nanotube to intensify the photocatalytic activity and surface hydrophilicity of TiO_2_. The introduction of PANI coated TiO_2_ nanotube enhanced the surface roughness and hydrophilicity properties of the nanocomposite membrane, which in turn resulted in the increase in pure water flux to 484 L m^−2^ h^−1^ from 312 L m^−2^ h^−1^ for a neat PVDF membrane while maintaining high rejection for methyl orange under UV light irradiation. The photocatalytic activity of Ti nanotubes rendered a self-cleaning feature to the nanocomposite membrane by photodegrading the dye molecule, hence mitigating organic fouling.

### 4.4. Discussion

As summarized in Table 1, significant progress has been made in developing TiO_2_-incorporated nanocomposite membranes. Great efforts have been made in exploring the potentials of multidimensional TiO_2_ nanostructures as modifying agents for a liquid separation membrane. Due to their simplicity and easily controlled reaction conditions, sol–gel and hydrothermal reactions have been widely used for the synthesis or the interconversion of multidimensional TiO_2_. It has been generally observed that, from integrally skinned asymmetric membranes to TFC membranes or from UF to RO, regardless of the membrane configuration and membrane processes, the incorporation of TiO_2_ has significantly contributed to improved hydrophilicity, which consequently resulted in enhanced water flux and antifouling properties. 1D and 2D TiO_2_ have been increasingly investigated owing to their structural uniqueness that is beneficial for the application of liquid separation membranes. Particularly, 1D TiO_2_ nanotubes offer large surface areas of hydrophilic moieties and hollowed structures that provide additional paths for water passage. On the other hand, sheet-like TiO_2_ nanofiller provides a short diffusion path distance for water transport in combination with large active sites and interfacial contact areas. However, the incorporation of 3D TiO_2_ in liquid separation nanocomposite membranes is still very limited, probably because of the difficulty in forming a defect-free membrane with the incorporation of 3D nanostructures. Based on the reported work, it is not surprising that ex situ blending and surface deposition are the two most-applied approaches for the preparation of TiO_2_-incorporated nanocomposite membranes. These approaches conveniently allow the post-synthesis surface functionalization of TiO_2_ not only to address the agglomeration issue at high nanofiller loading, but more importantly to add extra functionality to TiO_2_. One example is the coupling of TiO_2_ with other nanostructures such as Ag nanoparticles and GO to improve the photocatalytic activity. With the desired synergistic effects obtained from the hybrid system, the architecture of the TiO_2_-based hybrid has come to the forefront of the research in this area.

## 5. Conclusions and Outlook

Designing a membrane with high water permeability, selectivity, and antifouling properties is of primary importance in membrane-based liquid separation. The development of a nanocomposite membrane has undoubtedly provided a straightforward solution to address the current bottlenecks of commercial liquid-separation membranes. The foundational studies on the dimensionality of nanostructured materials have opened new avenues for the utilization of this material in liquid-separation membranes where the geometrical structure and dimension can be customized for application-specific requirements. The research efforts made to date have evidently shown that nanocomposite membrane is a promising tool to achieve better liquid-separation performances for various applications. Nevertheless, the identification and assessment of nanocomposite-membrane-related challenges are required to promote their practical applications on an industrial scale.

Based on recent exemplary studies, several significant research gaps and the corresponding research direction for the near future have been identified. Although TiO_2_ with various dimensions and morphology has been used for the fabrication of nanocomposite membranes, a direct comparison of the roles and functions of TiO_2_ with different structural properties in affecting the physico-chemical properties of the resultant nanocomposite membranes for liquid separation is still very limited. The comparison under the same membrane preparation procedure and filtration setting will be very meaningful to assess the differences in membrane properties and performances, hence providing more concrete evidence to justify the selection of nanomaterial. Optimizing the orientation and alignment of nanomaterials in a polymeric matrix allows the full utilization of the structural uniqueness of the nanostructures. For example, the water-channel-created 1D TiO_2_ nanotubes can only be materialized if the nanotubes are vertically oriented. Regrettably, the studies reported to date have not paid close attention to this subject matter. The use of external forces such as magnetic and electrical forces for nanomaterial alignment has been reported but has not received much attention mainly because of the complexity of the setup and the uncontrollable response of most nanomaterials toward the applied external forces. In view of the importance of this aspect, it is encouraged to explore and establish industrial-friendly approaches to enable a controlled orientation of nanomaterials in nanocomposite membranes.

Despite the remarkable research advancements, to date, nanocomposite membranes have not penetrated into the market; hence, the benefit has yet to achieve its initial promise. The development of nanocomposite membranes not only requires material innovation in terms of structural and compositional tailoring, but also calls for new processing and fabrication techniques that make it possible to develop nanocomposite membranes enabled by multidimensional nanomaterials. New and innovative synthesis methods should be explored to offer not only reproducible properties but their easy incorporation into a polymer matrix. For instance, more in situ nanomaterial formation and functionalization techniques other than sol–gel method should be explored as in situ formation allows better interaction between the nanomaterial and polymer chain, hence suppressing nanomaterial agglomeration tendency. The commercialization potential of nanocomposite membranes largely lies in the scalability and processibility of the materials. High-throughput manufacturing of multidimensional materials with high quality at affordable cost is a key toward realizing commercialization. In terms of material stability, the challenges in addressing trade-offs between long-term stability and high performance of nanocomposite membranes are highly desired in the pre-commercialization stage. The current challenges in reproducibility call for the exploration of new approaches with better reproducibility and reliability. The advances made in additive manufacturing offer an opportunity for 3D printing, which enables remarkable freedom for the design and customization of the nanomaterials and the nanocomposite membranes. Not only can the material issues such reproducibility be addressed, the 3D approach also allows rapid prototyping and on-site, on-demand modifications [140]. Machine learning is a helpful tool for predicting the properties of multidimensional nanomaterials, especially when traditional methods are too expensive or time-consuming to be implemented.

The geometrical uniqueness of nanomaterials for liquid-separation-membrane performance enhancement can be further exploited through a membrane templating technique. Unlike typical nanocomposite membranes in which the nanomaterials are embedded within the membrane matrix or on the membrane surface, the membrane templating technique involves the incorporation of nanomaterials during membrane fabrication, followed by their removal from the as-fabricated membrane with spaces imprinted with the structures of the nanomaterials removed. Templating based on a pre-existing guide nanomaterial with desired structure and nanoscale features is a powerful technique for the controlled formation of membrane structure as this technique allows the formation of membrane porous structure into forms and connectivity that are otherwise hard to acquire. Molecularly templated membranes can offer fast water permeability and solute selectivity when a nanostructured templating agent with the right geometry and structures is used. To date, several types of nanomaterials have been attempted as templating agents for membranes. The dimensions and structures of the nano-templating agent are the most crucial factor in constructing a porous structure. Compared to spherical nanoparticles, a membrane matrix imprinted with 1D nanorods has been shown to possess hollowed structures and channels that are highly interconnected to promote water flow [141]. As templated membranes have been increasingly studied, it is interesting to pay more attention to comparing the performance of liquid-separation membranes templated with nanostructures of the same chemical composite but different structural geometries. Such studies allow an optimum tailoring of membrane morphological and textural properties.

The development of nanocomposite membranes using multidimensional nanostructures presents tremendous research opportunities for the next several decades. In this review, based on TiO_2_ nanostructures, the unique features of nanomaterials with multidimensional geometries have been discussed and the corresponding roles of these nanostructures in a nanocomposite membrane for liquid-separation applications have been interpreted. It is concluded that the incorporation of TiO_2_ as a surface-modifying agent or nanofiller has played a significant role in countering the limitations of currently available liquid-separation membranes. Although TiO_2_ serves as a classic example of a nanomaterial used for the development of a nanocomposite membrane, the insights provided in this review are not only applicable to TiO_2_ but also to other potential nanostructures that can be prepared in multidimensions and geometries. Lastly, while nanocomposite membranes offer plenty of exciting research and commercialization opportunities, the major technological challenges related to their production must be addressed to advance the material for practical application.

## Figures and Tables

**Figure 1 nanomaterials-13-00448-f001:**
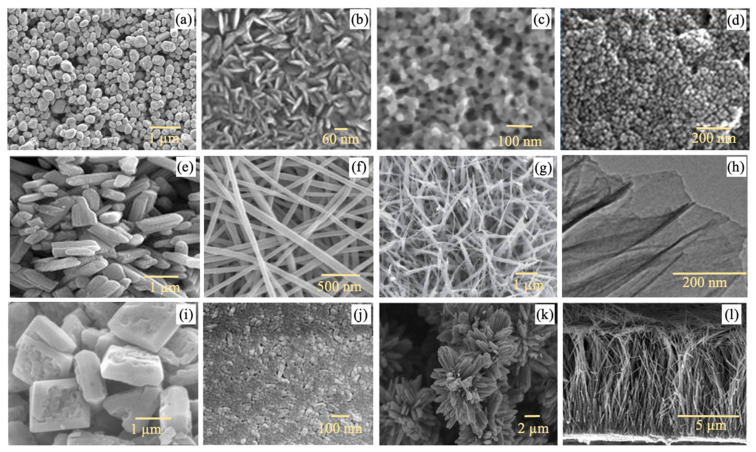
FESEM images of TiO_2_ nanostructures with different sizes and shapes: (**a**) nanospheres [52], (**b**) nanoplatelets [53], (**c**) nanopores [54], (**d**) quantum dots [55], (**e**) nanorods [57], (**f**) nanofibers [58], (**g**) nanowires [59], (**h**) nanosheets [65], (**i**) nanoplates [66], (**j**) nanofilms [66], (**k**) nanoflowers [69], and (**l**) nanoforest structures [70]. Reprinted with permission.

**Figure 2 nanomaterials-13-00448-f002:**
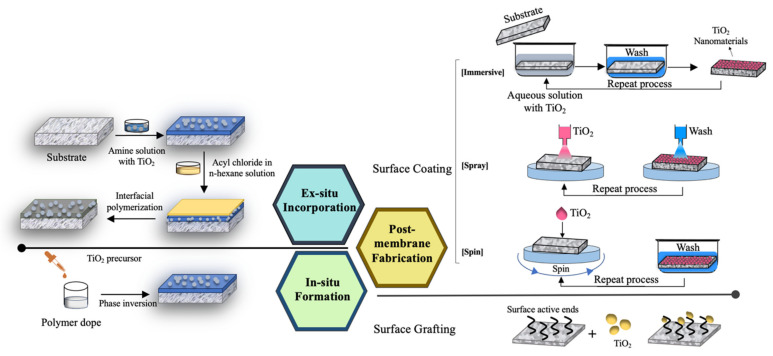
Fabrication of nanocomposite membrane via ex situ incorporation, in situ formation, and post-membrane fabrication surface coating or grafting.

**Figure 3 nanomaterials-13-00448-f003:**
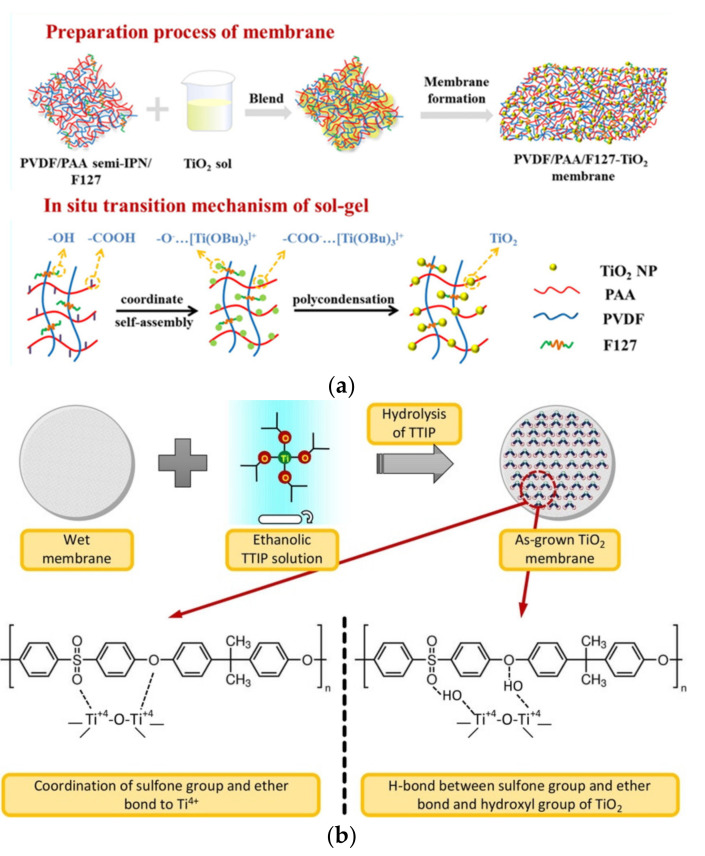
Schematic illustration of (**a**) the blending of TiO_2_ sol with polymer dope and the in situ transition mechanism of sol–gel to form TiO_2_ nanoparticles [116] and (**b**) the in situ hydrolysis of TiO_2_ precursor on the polymer backbone [118]. Reprinted with permission.

**Figure 4 nanomaterials-13-00448-f004:**
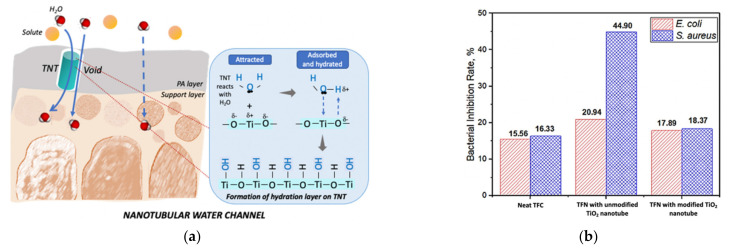
(**a**) Schematic illustration of surface hydroxyl groups and nanochannels created by TiO_2_ nanotubes in a polyamide layer of an RO TFC membrane [127], (**b**) bacterial inhibition efficiency of neat and nanocomposite TFC membrane against *E. coli* and *S. aureus* [128]. Reprinted with permission.

**Figure 5 nanomaterials-13-00448-f005:**
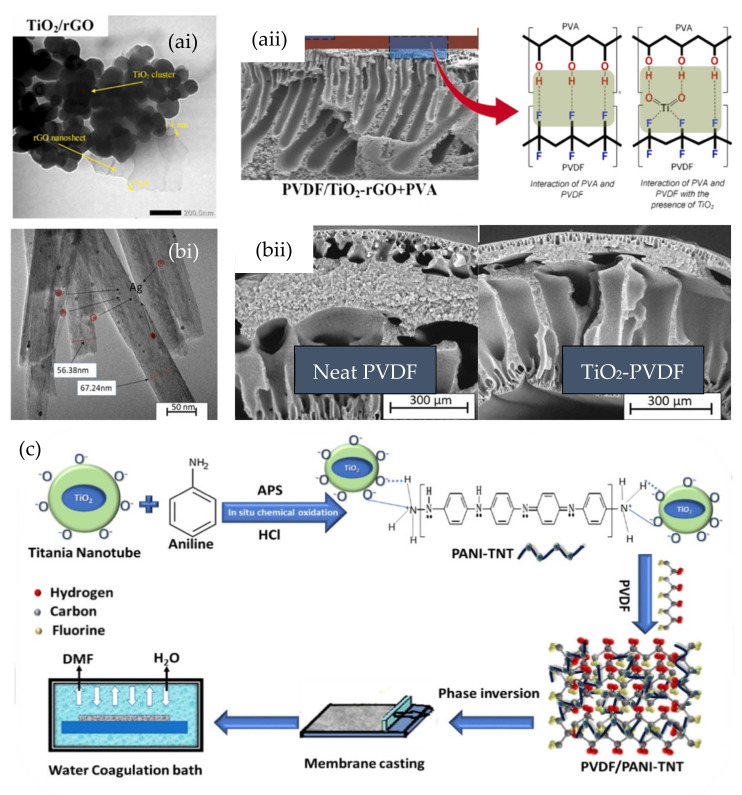
(**ai**) Morphology of rGO/TiO_2_: (**aii**) interaction between PVA−TiO_2_−PVDF in nanocomposite membrane [133]; (**bi**) surface morphology of Ag−doped TiO_2_ nanotubes; (**bii**) cross−sectional morphology of neat dual−layer PVDF membrane and nanocomposite membrane incorporated with Ag−doped TiO_2_ nanotubes [136]; (**c**) schematic flow of the preparation of PVDF nanocomposite inorporated with PANI−functionalized TiO_2_ nanotube [137]. Reprinted with permission.

**Table 1 nanomaterials-13-00448-t001:** Exemplary liquid separation nanocomposite membrane incorporated with multidimensional TiO_2_.

Base Polymer	TiO_2_ Geometry	Technique	Application	Membrane Process	Improved Parameters	Ref.
PVDF/PAA	Spherical	In situ surface growth	Protein, oil/water separation	UF	Water flux, antifouling	[116]
PES	Spherical	Surface deposition	Dye/salt separation	NF	Water flux, dye removal, antifouling	[123]
PVDF	Spherical	Surface deposition	Dye removal	Photocatalytic [119]	Dye removal	[119]
PVDF	Spherical	Surface deposition	Dye removal	Photocatalytic	Dye adsorption–degradation	[120]
Polyurethane/cellulose acetate	Spherical	Ex situ blending	-	Photocatalytic	Antimicrobial	[106]
PVDF	Spherical	Surface deposition	Protein separation	UF	Water flux, BSA rejection, antimicrobial (*E. coli*)	[132]
PVDF	Spherical	Ex situ blending	Dye separation	Photocatalytic	Water flux, COD removal, self-cleaning	[133]
PVDF	Spherical	Surface deposition	Phenol removal	Photocatalytic	Phenol removal	
PA TFC	Nanotube	Ex situ blending during IP	EDC removal	RO	Water flux, antifouling	[127]
PA TFC	Nanotube	Ex situ blending during IP	Desalination	NF	Water flux, antifouling, antimicrobial (*E. coli*, *S. aureus*)	[128]
PVDF	Nanotube	Ex situ blending	Dye separation	Photocatalytic	Water flux, antifouling	[137]
PVDF	Nanotube	Ex situ blending	Color removal	Photocatalytic	Water flux, antifouling	[138]
PVDF	Nanotube	Ex situ blending	Color removal	Photocatalytic	Water flux, antifouling, antimicrobial (*P. aeruginosa*)	[136]
PA TFC	Nanosheet	Surface deposition	Desalination	RO	Water flux, antifouling, NaCl rejection	[121]
PA TFC	Nanosheet	Surface deposition	Oil/water separation	RO	Water flux, antifouling, oil rejection	[129]
PA TFC	Nanosheet	Surface deposition	Desalination	RO	Water flux, antifouling, NaCl rejection	[130]

## Data Availability

Not applicable.

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
