# Peer review of "Modification of Liquid Separation Membranes Using Multidimensional Nanomaterials: Revealing the Roles of Dimension Based on Classical Titanium Dioxide"

_nanomaterials, 2023, doi:10.3390/nano13030448_

Round 1
Reviewer 1 Report
It is an excellent review article on the state-of-the-art modifications of liquid separation membrane using TiO2 as an example of multidimensional nanomaterials. The performances of TiO2-incorporated nanocomposite membranes are discussed based on the special features rendered by their structure and dimensions. The paper is well organized and an excellent review paper in the liquid separation membranes.
Author Response
It is an excellent review article on the state-of-the-art modifications of liquid separation membrane using TiO2 as an example of multidimensional nanomaterials. The performances of TiO2-incorporated nanocomposite membranes are discussed based on the special features rendered by their structure and dimensions. The paper is well organized and an excellent review paper in the liquid separation membranes.
Response: Thank you for the positive comments
Reviewer 2 Report
The authors have presented a review of nanocomposite membranes incorporating titanium dioxide, with a specific focus on “the roles of dimension”. I appreciate the authors choice to focus on this aspect of nanocomposite membranes given how many reviews have appeared on the general topic of nanocomposite membrane materials over the past few years. However, the review itself actually has very little discussion of this topic – the only sections that specifically discuss the impact of dimensionality of the performance of different membranes are sections 4.1 (which examines one-dimensional TiO2) and 4.2 (which examines two-dimensional TiO2). The rest of the manuscript provides a somewhat “rambling” discussion of TiO2 materials and their properties.
In addition to this general concern, I had a number of other more specific comments:
1. The authors describe a large number of morphologies / structures of TiO2, but they do not provide clear descriptions of how some of these structures differ. For example, what is the difference between a nanorod, nanofiber, and nanowire? Similarly, how do nanospheres and quantum dots differ? The authors need to clearly define the key morphological / size properties that distinguish these materials to help the readers understand the points that are being made in the text.
2. The authors need to provide a clearer discussion of the very different transport behavior of RO / NF vs UF / MF membranes. For example, TiO2 nanotubes may be of interest in RO or NF membranes where one may be able to get significant water transport through the nanotubes (while rejecting other solutes). However, this would be a significant disadvantage in most UF / MF applications where one would want the small solutes to pass through the membrane. This is never clearly described in the manuscript.
3. The text has a number of errors / poor word choice throughout. As some examples:
Line 13 – “the merge of nanotechnology and membrane technology” “the merger”
Line 133 – “Membrane modification through in the introduction” “in” should be deleted
Line 516 – “Section 4.2. Two-dimensional TiO” this should obviously be “TiO2”
Line 703 – “hence the benefit of has” “of” should be deleted
Line 718 – “reproducibility of huge room” I am not sure what was intended here
Author Response
Response to Reviewer 2
The authors have presented a review of nanocomposite membranes incorporating titanium dioxide, with a specific focus on “the roles of dimension”. I appreciate the authors choice to focus on this aspect of nanocomposite membranes given how many reviews have appeared on the general topic of nanocomposite membrane materials over the past few years. However, the review itself actually has very little discussion of this topic – the only sections that specifically discuss the impact of dimensionality of the performance of different membranes are sections 4.1 (which examines one-dimensional TiO2) and 4.2 (which examines two-dimensional TiO2). The rest of the manuscript provides a somewhat “rambling” discussion of TiO2 materials and their properties.
Response:
Thank you for the comments. In our opinion, before elaborating the impacts of dimensionality of TiO2 on the properties and separation performances, the discussion on the features of multidimensional TiO2, as presented in Section 2 is important to correlate the different roles played by 0D-3D TiO2 in membrane application. For example, 1D and 2D TiO2 have been associated to water transport facilitation, but this feature is unmet by 0D TiO2.
Similarly, Section 3 provides insights into different fabrication techniques of nanocomposite membranes, in which some of these are closely related to the dimensionality of TiO2. For example, in-situ fabrication of nanocomposite membrane is applicable to 0D TiO2 but not favourable for 1D and 2D TiO2 due to the more complex synthesis route. Another example, the incorporation of vertical 1D TiO2 and 3D in the selective layer of nanocomposite membrane is not favourable as they may cause destruction to the selective layer.
Therefore, in our opinion, these sections are important to provide better understanding on the roles of multidimensional in membrane fabrication and applications.
In addition to this general concern, I had a number of other more specific comments:
- The authors describe a large number of morphologies / structures of TiO2, but they do not provide clear descriptions of how some of these structures differ. For example, what is the difference between a nanorod, nanofiber, and nanowire? Similarly, how do nanospheres and quantum dots differ? The authors need to clearly define the key morphological / size properties that distinguish these materials to help the readers understand the points that are being made in the text.
Response:
The synthesis and structures of multidimensional TiO2 are not the major focuses of this review manuscript. In our opinion, the detailed descriptions of the differences among these structures tend to add “rambling” contents to this review.
However, appreciating the comment raised by the reviewer, references related to the syntheses and modifications, morphologies and structural properties as well as applications have been added to guide the interested readers for further reading. (Page 4, Section 2)
- The authors need to provide a clearer discussion of the very different transport behavior of RO / NF vs UF / MF membranes. For example, TiO2nanotubes may be of interest in RO or NF membranes where one may be able to get significant water transport through the nanotubes (while rejecting other solutes). However, this would be a significant disadvantage in most UF / MF applications where one would want the small solutes to pass through the membrane. This is never clearly described in the manuscript.
Response:
Despite the differences in the transport behaviours of RO/NF and UF/MF, TiO2 incorporated nanocomposite membranes (integrally skinned asymmetric or thin film composite) have been used for these processes to improve water flux, antifouling, self-cleaning and anti-microbial properties. The justifications of applying TiO2 in these membranes have been discussed in Section 2.
Based on our understanding and knowledge in this field, although solute rejection (for eg, through size exclusion) through the tubular structure of TiO2 has not been reported, although theoretically, the lumen size of tubular nanostructures can be tailored for ion-gating. Therefore, we believe that, regardless of the types of membrane processes, TiO2 can be advantageously used to improve membrane properties based on the abovementioned features.
- The text has a number of errors / poor word choice throughout. As some examples:
Line 13 – “the merge of nanotechnology and membrane technology” à “the merger”
Line 133 – “Membrane modification through in the introduction” à “in” should be deleted
Line 516 – “Section 4.2. Two-dimensional TiO” à this should obviously be “TiO2”
Line 703 – “hence the benefit of has” à “of” should be deleted
Line 718 – “reproducibility of huge room” à I am not sure what was intended here
Response:
The errors have been corrected. The manuscript has been rechecked to minimize grammatical/typos errors in the manuscripts.
Reviewer 3 Report
This manuscript summarized the state-of-the-art modifications of liquid separation membranes using TiO2 as a classical example of multidimensional nanomaterials. The performances of TiO2-incorporated nanocomposite membranes are discussed with attentions placed on the special features rendered by their structure and dimensions. Given the importance of liquid separation membrane and the development of TiO2-based nanoparticles, the referee would imagine the manuscript may attract broad interests. However, the referee will not recommend publication in nanomaterials in its current state unless the following issues are considered.
1. Line 69. The limitations of conventional membranes should be described specifically.
2. Line 264, This paragraph mentioned the antibacterial properties after introducing TiO2 but did not discuss the liquid separation performance of membranes.
3. Line 294. The “EDC” should be explained when emerged for the first time.
4. Section 4.2. The title “TiO” should be corrected to “TiO2.”
5. Figure 5 should be optimized with unified marks.
6. Line 636. The unit format of water flux should be corrected.
7. Line 650. TiO2 1D and 2D TiO2 should be unified.
Author Response
Response to Reviewer 3
This manuscript summarized the state-of-the-art modifications of liquid separation membranes using TiO2 as a classical example of multidimensional nanomaterials. The performances of TiO2-incorporated nanocomposite membranes are discussed with attentions placed on the special features rendered by their structure and dimensions. Given the importance of liquid separation membrane and the development of TiO2-based nanoparticles, the referee would imagine the manuscript may attract broad interests. However, the referee will not recommend publication in nanomaterials in its current state unless the following issues are considered.
Thank you for the comments and feedback. The changes made are marked in green in the revised manuscript.
- Line 69. The limitations of conventional membranes should be described specifically.
Response:
The limitations of conventional membranes have been described in Line 87-90, Line 250-252.
The drawbacks of conventional membranes derived from commercial polymers especially in terms of permeability-rejection trade off and high fouling propensity can be address with the new properties introduced by the nanomaterials.
Many polymeric membranes used for liquid separation are commercially available. However, due to their hydrophobicity, polymeric membranes such as PSF and PVDF are prone to membrane fouling.
- Line 264, This paragraph mentioned the antibacterial properties after introducing TiO2but did not discuss the liquid separation performance of membranes.
The antibacterial property of TiO2 has no direct impact on the separation performance but it can be effectively used to suppress biofouling of membrane. The related description has been provided in Line 273-277
The introduction of TiO2 as biocidal agent in liquid nanocomposites membrane has enabled new antibiofouling function during liquid separation. Interestingly, the TiO2 incorporated nanocomposites membrane can exert a non-contact biocidal action to achieve disinfection capabilities, thus the release of potentially toxic nanoparticles is not of major concern.
- Line 294. The “EDC” should be explained when emerged for the first time.
The full name of EDC has been provided: endocrine-disrupting compounds (EDCs)
- Section 4.2. The title “TiO” should be corrected to “TiO”
All TiO has been corrected to TiO2
- Figure 5 should be optimized with unified marks.
We are unsure about this comment. However, the layout of Figure 5 has been improved
- Line 636. The unit format of water flux should be corrected.
The unit of water flux has been corrected
- Line 650. TiO2 1D and 2D TiO2should be unified.
It has been corrected as 1D and 2D TiO2
Reviewer 4 Report
This review provides an overview and comments on the state-of-the-art modifications of liquid separation membrane using TiO2 as a classical example of multidimensional nanomaterials. The performances of TiO2-incorporated nanocomposite membranes are discussed with attentions placed on the special features rendered by their structure and dimensions. The innovations and breakthrough made in the synthesis and modifications of structure controlled TiO2 and its composites have enabled fascinating and advantageous properties for development of high-performance nanocomposite membranes for liquid separation. Meanover, the manuscript is comprehensive. In addition, Nidal Hilal is one of my favorite professors. Therefore, the manuscript should be accepted. However, I have some questions.
1. Maybe Section 2 and 3 can be combined.
2. Miswrite the word TiO2 to TiO in some place.
3. Section 5 is looks like Introduction.
Author Response
Response to Reviewer 4
This review provides an overview and comments on the state-of-the-art modifications of liquid separation membrane using TiO2 as a classical example of multidimensional nanomaterials. The performances of TiO2-incorporated nanocomposite membranes are discussed with attentions placed on the special features rendered by their structure and dimensions. The innovations and breakthrough made in the synthesis and modifications of structure controlled TiO2 and its composites have enabled fascinating and advantageous properties for development of high-performance nanocomposite membranes for liquid separation. Meanover, the manuscript is comprehensive. In addition, Nidal Hilal is one of my favorite professors. Therefore, the manuscript should be accepted. However, I have some questions.
- Maybe Section 2 and 3 can be combined.
Response: Thank for the comments. In our opinion, Section 2 and Section 3 serve different purposes; the former discusses the features of multidimensional TiO2 and the latter elaborates the fabrication techniques of nanocomposite membranes.
- Miswrite the word TiO2 to TiO in some place.
Response: All TiO has been corrected to TiO2
- Section 5 is looks like Introduction.
Paragraph 1 of section 5 has been condensed to omit some general statements. The remaining contents focus on the challenges in developing nanocomposite membranes and suggestions for future studies. The last paragraph concludes the manuscript.
Round 2
Reviewer 2 Report
The authors have appropriately revised the paper to correct the minor errors noted in my original review. However, their response to the more substantive concerns is completely inadequate. In particular:
1. I had requested that the authors explain the difference between a nanosphere and a quantum dot as well as between a nanorod, nanofiber, and nanowire. The authors indicate in written their response that “detailed descriptions of the differences among these structures tend to add “rambling” contents to this review.” First, I did not ask for a detailed description. Second, I completely disagree that defining the difference between a nanosphere and a quantum dot requires “rambling”. Since this paper is specifically focused on nanomaterial structures, providing the readers with a brief definition of these terms in the context of this Review is, in my opinion, essential.
2. Beginning in Line 167 of the revised paper, the authors state that “this review primarily aims to identify the knowledge gaps in the design of a high-performance nanocomposite membrane through the selection of TiO2 nanomaterials with the right material dimension and structures.” As I noted in my original review, I felt that the authors did not follow through on this stated goal very effectively. Their response to this comment was “The synthesis and structures of multidimensional TiO2 are not the major focuses of this review manuscript.” This response is, in my view, inconsistent with the authors’ stated objective for this Review paper.
3. In addition to the above, I feel that the authors’ discussion of the published literature is in some places inadequate. For example, Reference [113] is shown as the first entry in Table 1 – the authors indicate that this work describes a “Photocatalytic” membrane process. However, the paper itself describes the development of a composite ultrafiltration membrane incorporating TiO2 gel nanoparticles that has high water flux and good BSA rejection. The paper also shows that this composite membrane is effective in treating water containing humic acid and oil-water emulsions. The paper does describe the use of UV irradiation to restore the water permeability after fouling, but it is very misleading to refer to this work as focusing on a “Photocatalytic membrane process.”
Author Response
The authors have appropriately revised the paper to correct the minor errors noted in my original review. However, their response to the more substantive concerns is completely inadequate. In particular:
Thank you for the additional comments raised by the reviewer. The manuscript has been revised accordingly.
- I had requested that the authors explain the difference between a nanosphere and a quantum dot as well as between a nanorod, nanofiber, and nanowire. The authors indicate in written their response that “detailed descriptions of the differences among these structures tend to add “rambling” contents to this review.” First, I did not ask for a detailed description. Second, I completely disagree that defining the difference between a nanosphere and a quantum dot requires “rambling”. Since this paper is specifically focused on nanomaterial structures, providing the readers with a brief definition of these terms in the context of this Review is, in my opinion, essential.
The description related to the difference in these nanostructures is provided. Refer to Line 196-211.
- Beginning in Line 167 of the revised paper, the authors state that “this review primarily aims to identify the knowledge gaps in the design of a high-performance nanocomposite membrane through the selection of TiO2 nanomaterials with the right material dimension and structures.” As I noted in my original review, I felt that the authors did not follow through on this stated goal very effectively. Their response to this comment was “The synthesis and structures of multidimensional TiO2are not the major focuses of this review manuscript.” This response is, in my view, inconsistent with the authors’ stated objective for this Review paper.
The statement: ‘….the selection of TiO2 nanomaterials with the right material dimension and structures’ refers to the intentions of the authors to correlate the dimension, morphology and structures of TiO2 nanomaterials with the intended applications and the membrane fabrication methods. Therefore, the synthesis methods of these TiO2 (hydrothermal, sol-gel and etc) are not discussed in detailed in this manuscript.
- In addition to the above, I feel that the authors’ discussion of the published literature is in some places inadequate. For example, Reference [113] is shown as the first entry in Table 1 – the authors indicate that this work describes a “Photocatalytic” membrane process. However, the paper itself describes the development of a composite ultrafiltration membrane incorporating TiO2 gel nanoparticles that has high water flux and good BSA rejection. The paper also shows that this composite membrane is effective in treating water containing humic acid and oil-water emulsions. The paper does describe the use of UV irradiation to restore the water permeability after fouling, but it is very misleading to refer to this work as focusing on a “Photocatalytic membrane process.”
The process of this entry (116 in the revised manuscript) has been changed to ‘UF’
Reviewer 3 Report
The authors have done an acceptable job in addressing the reviewers' comments and revising the manuscript.
Author Response
The authors have appropriately revised the paper to correct the minor errors noted in my original review. However, their response to the more substantive concerns is completely inadequate. In particular: Thank you for the additional comments raised by the reviewer. The manuscript has been revised accordingly. 1. I had requested that the authors explain the difference between a nanosphere and a quantum dot as well as between a nanorod, nanofiber, and nanowire. The authors indicate in written their response that “detailed descriptions of the differences among these structures tend to add “rambling” contents to this review.” First, I did not ask for a detailed description. Second, I completely disagree that defining the difference between a nanosphere and a quantum dot requires “rambling”. Since this paper is specifically focused on nanomaterial structures, providing the readers with a brief definition of these terms in the context of this Review is, in my opinion, essential. The description related to the difference in these nanostructures is provided. Refer to Line 196-211. 2. Beginning in Line 167 of the revised paper, the authors state that “this review primarily aims to identify the knowledge gaps in the design of a high-performance nanocomposite membrane through the selection of TiO2 nanomaterials with the right material dimension and structures.” As I noted in my original review, I felt that the authors did not follow through on this stated goal very effectively. Their response to this comment was “The synthesis and structures of multidimensional TiO2 are not the major focuses of this review manuscript.” This response is, in my view, inconsistent with the authors’ stated objective for this Review paper. The statement: ‘….the selection of TiO2 nanomaterials with the right material dimension and structures’ refers to the intentions of the authors to correlate the dimension, morphology and structures of TiO2 nanomaterials with the intended applications and the membrane fabrication methods. Therefore, the synthesis methods of these TiO2 (hydrothermal, sol-gel and etc) are not discussed in detailed in this manuscript. 3. In addition to the above, I feel that the authors’ discussion of the published literature is in some places inadequate. For example, Reference [113] is shown as the first entry in Table 1 – the authors indicate that this work describes a “Photocatalytic” membrane process. However, the paper itself describes the development of a composite ultrafiltration membrane incorporating TiO2 gel nanoparticles that has high water flux and good BSA rejection. The paper also shows that this composite membrane is effective in treating water containing humic acid and oil-water emulsions. The paper does describe the use of UV irradiation to restore the water permeability after fouling, but it is very misleading to refer to this work as focusing on a “Photocatalytic membrane process.” The process of this entry (116 in the revised manuscript) has been changed to ‘UF’